# A Cell Biologist’s View on APOL1: What We Know and What We Still Need to Address

**DOI:** 10.3390/cells14130960

**Published:** 2025-06-24

**Authors:** Verena Höffken, Daniela Anne Braun, Hermann Pavenstädt, Thomas Weide

**Affiliations:** Medical Clinic D, University Hospital Münster, 48147 Münster, Germany; verena.hoeffken@ukmuenster.de (V.H.); danielaanne.braun@ukmuenster.de (D.A.B.); hermann.pavenstaedt@ukmuenster.de (H.P.)

**Keywords:** APOL1, Apolipoprotein L1, APOL1-mediated kidney disease, AMKD, renal risk variants, nephrotoxicity, topology, ion pore

## Abstract

*APOL1* is the most recent member of the *APOL* gene family and is expressed exclusively in humans and a few higher primates. More than twenty years ago, it was discovered that APOL1 protects humans from infections by trypanosome subspecies that cause African sleeping sickness. Interestingly, by a co-evolutionary process between parasite and host, two APOL1 variants emerged, which, in addition to their trypanotoxic effects, are simultaneously associated with a significantly increased risk for various different kidney diseases, which are now summarized as APOL1-mediated kidney diseases (AMKDs). The aim of this review is to highlight and formulate key aspects of APOL1’s cell biologic features, including questions and unaddressed aspects. This perspective may contribute to a deeper understanding of APOL1-associated cytotoxicity as well as AMKDs.

## 1. Introduction

*APOL1* is the most recent member of the *APOL* gene family and is expressed only in humans and a few higher primates. Thus, in most mammals, the gene is absent and there is even a study suggesting that loss of the *APOL1* gene in humans may does not lead to severe or life-threatening consequences. This raises the interesting question why a relatively recent, non-essential (some might say dispensable) gene is of such great interest from an evolutionary, parasitological, virological, and nephrological perspective. The answer is that this “superfluous” *APOL1* gene is linked to two severe human diseases: on one hand, protection from human African trypanosomiasis (HAT), also known as African sleeping sickness, and on the other hand, a range of kidney diseases now collectively referred as APOL1-mediated kidney diseases (AMKD). The development of therapeutic strategies and causal treatments for both HAT and AMKD requires a deep understanding of the molecular and cell biologic functions of APOL1.

We begin by discussing aspects of the evolutionary background of the APOL family members and APOL1. In the next section, we will focus on the relationship between alternative splicing and membrane orientation and focus on intracellular localization of APOL1. Finally, we will discuss how APOL1 may disturb crucial cell biological processes which are currently considered as major driver of AMKDs at the cellular level.

## 2. Evolution of the *APOL* Gene Family

The evolution of the *APOL* gene family is based on a series of gene duplication events. In humans, all six *APOL* genes (*APOL1–APOL6*) are located in a cluster on chromosome 22 [1]. The origin of the *APOL* gene family can be traced back to *APOL6*, which represents the ancestral gene. Subsequent duplications result first in the *APOL5* gene, followed by four additional *APOL* genes called *APOL1-APOL4*. While *APOL6* (and *APOL5*) genes were found in a broad range of mammalian species, *APOL1–APOL4* are restricted to primates. Among them, *APOL1* is the most recent one and only expressed in humans and a few higher primates, such as gorillas and baboons [2,3].

Recently, the full genomic sequences of the human *APOL1* trough *APOL6* genes were aligned with each other to look for phylogenetically informative transposed elements. These elements, once inserted into the genome, are inherited by all descendant gene copies and species, making them useful markers for determining evolutionary relationships. This approach uncovered two primate-specific diagnostic SINEs (short interspersed nuclear elements) within the *APOL* gene cluster: *Alu*J and *Alu*Y. The *Alu*J element was found in the same intronic position of *APOL1–4* across all investigated primates, suggesting insertion in a common ancestor approximately 70 million years ago. In contrast, the *Alu*Y element was shared only between *APOL1* and *APOL2* and is present in old world monkeys and apes (Catarrhine primates) that diverged around 30 million years ago. These findings support the close evolutionary relationship between *APOL1* and *APOL2* and provide an independent, sequence-based perspective on the timing and pattern of *APOL* gene diversification [3]. Interestingly, evolutionary analyses also revealed that mice (rodents) show a distinct *APOL* gene diversification, which evolved independently from the primate lineage, leading to a total of twelve *APOL* genes, all of them clustered on chromosome 15 [3].

The evolutionary origin of *APOL6* can be traced back 300 million years to a gene known as *APOLD1* (Apolipoprotein L domain containing 1), which is also referred to as *Verge* (from: vascular early response gene). This gene is already present in birds, reptiles and mammals (*sauropsida* and *mammalia*), and is located outside of the murine and human *APOL* gene clusters as it localizes on chromosome 6 in mice and chromosome 12 in humans [3]. Thus, the diversification of the *APOL* gene family occurred largely independently in rodents and primates. This suggests that at least the genes *APOL1-4* have biological functions, which are largely specific or even unique to primates [3].

Already initial studies on human *APOL* genes have shown that their expression can be induced by inflammatory or immune triggers [2]. Remarkably, this pattern is also observed for the ancestral *APOLD1* gene, which shares moderate homology (~30% identity) at the protein level with the central region of human *APOL6*—specifically the Bcl-2 homology 3 (BH3) domain-like sequence motifs (BSMs) and membrane insertion module (MID) [4]. The *APOLD1* sequence is highly conserved, showing high similarity with the mouse *Apold1* and even the chicken ortholog. *APOLD1*, is mainly expressed in endothelial cells [4,5]. *Apold1* knockout mice showed that loss of this protein does not impair vascular development, but that it functions in maintaining angiogenic homeostasis. Moreover, animals lacking *Apold1* are more prone to thrombosis and in ischemia damaged tissue, indicating that repair of vascularity system is impaired [6,7,8,9]. This suggests that increased expression of the *Apold1* gene in response to inflammation, which follows ischemic injury, may play a role for improving neovascularization and thus, tissue repair. Together, the studies revealed that beyond sequence homology at the DNA and protein level that inducibility by immunological factors could be an additional evolutionary conserved feature within the APOL family.

Among all *APOL* family members, *APOL1* is by far the best studied gene. It was identified in 2003 as a trypanolytic factor that protects humans against infections with *Trypanosoma* blood parasites, the causative agents of HAT, also known as African sleeping sickness [10,11,12]. Pays and colleagues defined three functional domains in APOL1: a so-called pore-forming domain (PFD), which resembles a structure of bacterial toxin Colicin A that can form pores in membranes via a membrane insertion module. Within the PFD, APOL proteins contain also a BSM, that is associated with programmed cell death (PCD). The PFD is followed by membrane-addressing domain (MAD), which enables or facilitates membrane targeting and binding. In *Trypanosoma brucei rhodesiense* parasites resistant to APOL1, this resistance is mediated by the trypanosomal serum-resistance-associated (SRA) protein, which binds specifically to the C-terminal SRA-interacting domain (SID) of APOL1 [11,13] (Figure 1). Other members of the APOL family show a similar PFD-MAD-SID domain architecture. Based on this, Smith and Malik proposed in 2009 that the *APOL* gene family associated with cell death and immune defense has evolved, particularly at discrete sites of host–pathogen interaction [2,13].

APOL1 is unique within the APOL family as it can be released into the serum. This secretion is mediated by an N-terminal signal peptide (SP), that is only present in some but not all APOL1 splice variants (see below) and absent in almost all other APOL family members. This demonstrates that the cellular functions of the evolutionary older family members (APOL2-6) are linked to intracellular functions, most likely as part of the innate immunity [2,3]. This suggests, that the mainly extracellular localization of APOL1 may has been associated with different evolutionary selection processes in the context of host-pathogen interactions [2,3]. Indeed, APOL1 has been part of an evolutionary arms race between host and blood parasites. Two subspecies of *Trypanosoma brucei*—*T. b. rhodesiense* and *T. b. gambiense*—have evolved resistance to APOL1 by producing different forms of their main surface protein, called VSG (Variant Surface Glycoprotein). These forms are known as SRA in *T. b. rhodesiense* and TgsGP (*T. gambiense*-specific Glycoprotein) in *T. b. gambiense*. SRA directly binds to the C-terminal part of APOL1, blocking its function, while TgsGP decreases membrane fluidity of the parasite, which prevents APOL1 insertion. In response, two APOL1 sequence variants called G1 (S342G/I384M) and G2 (Δ388N, Δ389Y) emerged in humans of Sub-Saharan African ancestry approximately 10,000 to 20,000 years ago. These changes within the C-terminal region avoid SRA binding and restore resistance to *T. b. rhodesiense* [12,14,15,16] Remarkably, the presence of two copies of these variants (G1/G1, G2/G2, or G1/G2) on both alleles is associated with a significantly increased risk of a group of kidney diseases now referred to as AMKD [17,18]. Therefore, these variants are referred to as renal risk variants, RRVs.

Together, the *APOL* genes arose from duplications and have rapidly diversified. In addition to their high sequence homology and a highly conserved membrane-insertion or—association module, a key feature of this gene family is their conditional expression in response to inflammatory triggers (inducibility).

Open questions: While members of the *APOL* gene family are thought to be part of the innate immune system, their original and probably common cellular functions remain unclear. Expression patterns in tissues and immune cells are not well understood. Also, the mechanism of APOL induction (for example common responsive or regulatory elements) requires further investigation. Key questions include whether the BSMs have anti- or pro-apoptotic roles. It is also unclear to which extent the individual members of the APOL family have diverse or overlapping functions during inflammatory responses, and which evolutionary mechanisms have forced the evolution of different APOL family members (diversification).

## 3. APOL1: Splice Variants and Their Membrane Topology

APOL1—in contrast to its closest relative APOL2—is expressed in different splice variants through the exclusion of exon 2 and/or exon 4 out of the seven exons [2,3,19,20,21]. APOL1 variant A (vA) is encoded by exons 1 and 3–7 and represents the predominantly expressed splice isoform of APOL1 [20,21]. When all exons are utilized, variant B1 (vB1) is formed; variant B3 (vB3) is expressed when only exon 4 is excluded, whereas the splice variant C (vC) is expressed in case that both, exons 2 and exon 4 are absent [19].

The orientation of APOL1 is determined by exons 2–4, which encode a putative SP. This SP and consequently the orientation is essential for APOL1’s insertion into the ER lumen and its subsequent transport to the plasma membrane and/or release into the extracellular medium via the secretory pathway. Since it was discovered as a trypanolytic factor, APOL1 has also been seen as a protein that is associated with or binds to membranes. Indeed, this is the key observation why concepts of the APOL1 domain architecture suggested functional PFD and MAD, which can clearly be distinguished from the SRA-interacting module [11,13]. More recent work suggests that the SP-bearing splice variants of APOL1 (especially the vA variant) can behave like integral membrane proteins [3,22,23,24,25,26]. However, it is still under debate and so far, it is not completely understood how many membrane-spanning domains are present in APOL1 at their different cellular targeting membranes.

An artificially introduced N-glycosylation tag, which becomes glycosylated only when located in the ER lumen, demonstrated that the C-terminal end of the APOL1 splice variants vA, vB1, and vC shows a luminal orientation. In contrast, variant vB3, which lacks exon 4 and thus a functional SP, exhibits an opposite topology, with the C-terminus facing the cytoplasmic side, similar to the SP-lacking APOL2 [3] (Figure 2a).

In support of this, in silico analysis, identified putative SPs only in the N-termini of vA, vB1 and vC but not in vB3, or APOL2 and secreted alkaline phosphatase (SEAP) fusion proteins in which the endogenous SEAP-SP was replaced by the N-terminal sequences of APOL1 splice variants were only secreted into the medium in case of the functioning SPs of vA, vB1, and vC, whereas the N-termini of splice variant vB3 or APOL2 did not show any SP activity. [3]. The data suggest first that APOL2 and vB3 have a different membrane topology than the SP-containing splice variants, and second, that SP-containing splice variants have either no transmembrane regions (TMRs) at all—which could be the case for secreted APOL1 pools—or an even number of TMRs, two or four (Figure 2b). The studies are in line with findings by Scales and colleagues, proposing a “two and a half-spanning loop” concept, where in addition to the SP, two TMR are present in the PFD region, and a region between MAD and SID that “dips” into the membrane of podocytes [22,27]. Experiments by Schaub et al. using a protein-wide cysteine scanning mutagenesis coupled with a cysteine accessibility assay indicate a model with four TMRs, with the 4th TMR becoming a pore lining region in planar bilayers [25]. Moreover, most scenarios in which APOL1 acts as ion pore at the plasma membrane (PM) are also based on the 4-TM-concept in which N- and C-termini of APOL1 show a luminal (at the ER) or extracellular (at the PM) orientation [23,25,28].

It is worth noting that the orientation of APOL1 in membranes is crucial for an understanding of potential interactions of APOL1 wildtypes or the RRVs with other proteins or lipids. Moreover, in the case of APOL1 being a transmembrane protein, the accessibility of potential interaction sites (regions or loops) also determines which interactions are possible.

Since the RRVs are localized on exon 7, all four splice variants can carry the amino acid variants for G1 or G2 [20,21]. However, whether all splice variants combined with RRV alleles can trigger cytotoxic effects is still not fully clarified. On the one hand, evidence suggests that APOL1 lacking the SP (exon 4-negative transcripts) are less cytotoxic than APOL1 variants containing a functional SP [19,20,29,30]. On the other hand, there are numerous experiments indicating that cytoplasmic oriented APOL1 (Figure 2a; “like APOL2”) can still be cytotoxic [3,31,32,33,34,35].

In support of this, several studies have proposed pathomechanisms that require APOL1 to be oriented toward the cytoplasm (at least for the SID), for example, if it binds to lipids localized to the cytoplasmic leaflet of intracellular membranes [35,36,37], if it interacts with the outer mitochondrial membrane [38] or the mitochondrial permeability transition pore, or if it binds intracellular cofactors [35,39,40].

Except for the well-defined protein composition of extracellular trypanolytic factors (TLF), which contains APOL1, and the extracellular tripartite complex containing suPAR (soluble urokinase plasminogen activator receptor), αvβ3 integrin, and APOL1, only few binding partners have been identified. These binding partners include APOL3, non-muscle myosin 2A (NM2A), prohibitin-2 (PHB2) and VAMP8 [34,39]. Strikingly, APOL3, NM2A, and PHB2 are cytoplasmic proteins, while VAMP8 is a membrane protein with relevant domains facing the cytoplasm, indicating that the SP-containing APOL1 isoforms do not bind to any of them. Additionally, using proteomic approaches, it has been described that APOL1 can be associated with a variety of proteins within the mitochondrial matrix, with RRVs tending to promote APOL1 oligomerization more than the wildtype [41]. APOL1 oligomerizations have also been postulated [42] or reported and may play an important role in the formation of ion pores or channels [30,41] (see below, Figure 2b). Identifying more putative interaction partners of APOL1, its different splice isoforms, and its family members may be key to a better understanding of APOL1´s diverse biological functions.

Open questions: So far, it remains unclear what factors control or trigger the splicing of the *APOL1* gene *in vivo*. Furthermore, it still needs to be determined whether ER luminal or cytoplasmic orientations are restricted to the ER and plasma membranes, or if they are also present on membranes of other cellular compartments. It also remains to be shown what the relative levels are in different tissues *in vivo*. Even if the main APOL1-associated cytotoxicity originates from the SP-containing APOL1 vA splice variants, it should be clarified to what extent the other variants contribute to or modify cell-damaging effects.

Finally, since most expression studies are based on *in vitro* or murine models, and the *in vivo* expression patterns in AMKD patients remain poorly characterized, it is essential to analyze expression profiles at the single-cell level in patient tissues to gain deeper insights into the underlying pathomechanisms.

## 4. Intracellular Localization of APOL1

A central question is whether the intracellular localization of APOL1 is also the site where RRVs exert their pathophysiological functions. A further question is to what extent the presence of multiple intracellular pools of APOL1 suggests that it might have several distinct compartment- or organelle-specific tasks. It is also possible that, despite the existence of different intracellular pools, APOL1 may not have any normal, defined intracellular function at all. In fact, this would align well with the observation that APOL1 is absent in many organisms, and its loss does not seem to have significant consequences for humans.

To address these aspects numerous studies were performed to identify the intracellular localizations of APOL1. Early cell culture studies identified APOL1 on autosomes or autophagosomes, where it was associated with programmed autophagic cell death [36,39,43]. Further experiments identified APOL1 on the PM, early and late endosomes (EE and LE) and autophagosomal vesicles [44] (Figure 3).

Notably, Beckerman and colleagues also uncovered that the RRVs predominantly accumulated in late endosomes and autophagosomes vesicles, while the APOL1 wildtype was predominantly localized on endosomal vesicles [44]. Lecordier et al. found APOL1 together with APOL3 associated with Golgi membranes [40]. APOL1 was also identified on mitochondria [31,38,40,45], including pools even observed within the mitochondrial matrix, where they may induce cell damaging via mitochondrial translocation and opening the mitochondrial permeability transition pores [41] (Figure 3).

Interestingly, Chun et al. found significant pools of the APOL1 wildtype on lipid droplets (LDs) in the cell, whereas the RRVs mainly remained at the membranes of the endoplasmic reticulum (ER) [37,46]. LDs are compartments primarily serving as storage for triglycerides and cholesterol esters and play a central role in lipid metabolism. They are also involved in processes such as lipid homeostasis, signaling, and cell stress responses usually formed by a complex process between the two leaflets of ER membranes and are—in contrast to mentioned compartments above—only surrounded by a single membrane [47,48] (Figure 3).

The largest and most complex intracellular membrane system is the ER. It forms highly branched so-called sheets and tubes that create a complex network throughout the cell [49]. Apart from its key functions for the lipid synthesis for cell membranes and central calcium storage, it is responsible for protein synthesis and folding for both secreted and membrane-bound proteins. Indeed, most studies, using electron microscopy, immunofluorescence studies with anti APOL1 antibodies, as well as live-cell imaging analyses identified the ER as the predominant compartment for APOL1 [3,22,27,31,50]. Since APOL1 is described as secreted (trypanolytic factor) or as a transmembrane protein (e.g., as ion pore), it is likely that APOL1 pools are found at the ER and Gogi membranes. Interestingly, APOL2—similar as APOL1—is mainly found at the ER as well [3,22]. Thus, APOL1 and its closest related family member APOL2 bind predominantly to ER membranes, albeit in different orientations, indicating that ER targeting is a common and probably conserved feature at least for these two members of the APOL family [3,22] (Figure 3).

In this context it is worth noting, that particularly the ER forms numerous close contacts between various intracellular membranes called “membrane contact sites (MCSs)” [49,51]. These MCSs associated the ER with endo-lysosomes, autophagosomes, mitochondria as well as with the PM. Moreover, during the last decade it became clear that these MCSs not only serve as physical membrane-membrane interfaces but also as hubs for various signaling, including calcium, reactive oxygen, or lipid signaling, as well as for processes such as autophagy, lipid metabolism, membrane dynamics and cellular stress responses [49,51] (Figure 4).

Unlike APOL2, APOL1 is also found at the cell surface, albeit in very small fractions. This has been observed by several different studies [22,23,24,28,30,44]. Currently, many researchers are focusing on this small surface pool, as this fraction might be crucial for the cytotoxic properties associated with APOL1 RRVs, possibly due to ion channel activity localized at the plasma membrane (see below).

Finally, APOL1 is also present extracellularly, as circulating pool in the blood serum. Indeed, this was the fraction of APOL1 that was identified at first, as a trypanolytic factor. In this context it is worth noting that the trypanolytic fraction of APOL1 can be detected in the serum even without inflammatory or immunological triggers [12]. However, from a cell biologic point of view, APOL1’s extracellular or circulating pools should be considered only for splice variants that contain a SP (vA, vB1 and probably also vC), because in most cases they are the prerequisite for secretory proteins.

Previous work identified the liver (hepatocytes) as the main source of these circulating APOL1 [52] pools. However, Cheng et al. made the interesting observation that APOL1 is poorly secreted *in vitro* in hepatocytes, even in the presence of chemical chaperones, but efficiently released into the serum in transgenic mice ectopically expressing human APOL1 [50]. This indicates that APOL1 secretion requires factors that are not present in cell lines, or that the release of APOL1 into the serum is by far more complex than initially assumed. Thus, the cell biologic details about APOL1’s release into the serum remain to be addressed.

In contrast to all other APOL proteins, which act intracellularly due to the absence of an SP, APOL1 also has extracellular functions. These functions may extend beyond its protective trypanolytic role and could potentially play a role as a cofactor in the pathogenesis of AMKDs. Reiser and colleagues for example described a tripartite complex, consisting of circulating APOL1, circulating suPAR and activated αvβ3 integrin [53]. Here the APOL1 RRVs showed a higher affinity for suPAR-activated αvβ3 integrin than the APOL1 wildtype. In conclusion, the authors propose that hyperactive αvβ3 integrin due to APOL1 RRV led to proteinuria and kidney disease in mice [53]. Yet, most studies suggest that intracellular APOL1 pools are primarily responsible for APOL1-associated cytotoxic effects of the risk variants (see below). The observation that in kidney transplantation the risk of allograft failure is increased in donors with APOL1 RRVs, but not in APOL1 RRV-positive recipients, further support the interpretation, that kidney-expressed rather than circulating RRV pools promote AMKD [17,18,54,55]. However, these findings do not fully exclude paracrine effects of secreted APOL1 pools, indicating that via an “APOL1-suPAR- αvβ3 integrin axis” RRVs may contribute to cell-damaging effects, possibly as an aggravating cofactor.

Open questions: APOL1 is found in various intracellular pools, however, it still needs to be shown under which conditions APOL1 is mainly secreted and under which circumstances, it predominantly remains in intracellular pools. Additionally, for the intracellular pools, it is unclear where and how the ratio between APOL1 on the cell surface and on intracellular membranes is regulated. A key question in this context will be to address whether an imbalance between APOL1 translation at the ER, its insertion into ER membranes, and its subsequent export to the cell surface might be responsible for the high intracellular APOL1 pools, particularly at ER membranes. There is also a gap in our understanding of the cell biological details of how APOL1 is incorporated into HDL particles and released by liver cells. Regarding the extracellular pools of APOL1, it remains to be shown how association and uptake and subsequent membrane insertion are regulated and whether these mechanisms are cell-type specific.

## 5. APOL1 and Cellular Injury

APOL1-associated cell damage and toxicity is a common phenomenon observed in various *in vitro* and *in vivo* model systems in which APOL1 RRVs are overexpressed. These toxic effects which reduce cell viability have not only been reported in mammalian cells, but also in evolutionary distant models such as yeast, insect cells, and mice [56,57,58,59,60]. These observations raise a fundamental question: Why do mutated variants of a protein that is almost exclusively found in humans and that, on a molecular level, mediates a highly specialized host-parasite interaction, cause toxic effects across such a wide range of model systems? The most plausible explanation is that APOL1-RRVs interfere with cellular processes that are both highly conserved and essential for survival in all eukaryotic cells, including yeast and insects as well as mammalian cells of non-human and human origin. If this is the case, a follow-up question naturally arises about which cellular pathways or processes this could be, and why the RRVs are particularly harmful in these contexts, while the wildtype form (G0), even when overexpressed, does not show such toxic effects?

### 5.1. APOL1 Cytoxicity: Loss-of-Protection or Gain-of-Cytotoxicity?

Above mentioned aspects are closely linked to the question whether APOL1-pathomechanisms involve loss-of-function or a gain-of-toxic-dysfunction mechanisms. The increased risk for renal diseases mediated by the APOL1 RRVs follows a recessive inheritance pattern, meaning that two risk alleles (G1/G1, G2/G2, or G1/G2) are required to confer increased susceptibility to kidney disease. In classical Mendelian genetics, such recessive inheritance is typically associated with loss-of-function phenotypes—caused by partial or total deficiency of the gene product´s normal physiological role. However, current evidence increasingly supports the idea that APOL1-linked cytotoxicity is not due to the loss or absence of physiological functions, but rather to the acquisition of novel, harmful properties caused by gain-of-dysfunction mechanisms [55].

Several lines of evidence support this interpretation: First, except for a single study [61], most research shows that cytotoxic effects are almost exclusively linked with the increased expression of APOL1 RRVs in an African background [30,62]. Second, the expression level requires a threshold to cause cytotoxic effects, as expression of only one risk allele—even in hemizygous conditions [60]—is not sufficient, to cause cell damaging effects in a transgenic mouse model. Finally, APOL1 is absent in almost all mammals (except for a few primates), suggesting that it does not have essential and evolutionarily conserved functions. In support of this: the complete absence of APOL1 in humans is not associated with a higher risk of kidney disease [63]. These observations leave two possible mechanisms: either a loss-of-protection function, meaning that in presence of one RRV-allele, the presence of one wildtype allele is required to prevent cytotoxicity, or gain-of-cytotoxicity *(*or gain-of-dysfunction*),* whereby the RRVs acquire novel toxic properties. The latter seems more plausible, as some evidence also indicates that high levels of (non-African) APOL1 wildtype do not rescue cells from RRV-induced toxicity [28,64]

### 5.2. APOL1 Cytoxicity Due to Disturbances of the Endolysosomal and Autophagic System

Research of the years elucidated a plethora of pathomechanisms for APOL1, of which many fulfil the above-mentioned criteria: affecting conserved and crucial function by a gain-of-dysfunction mechanism. A cellular pathomechanism, that has been frequently discussed in this context are disturbances of the endo-lysosomal pathway or the autophagy-lysosomal system, in a way that resembles its toxic mechanism in *c* parasites. This is an attractive idea, as it does not propose an entirely new function for APOL1, but rather a misdirected version of its known trypanotoxic activity. In this scenario, APOL1 forms pores in lysosomal membranes of eukaryotic cells (similar to trypanosomes), leading to lysosomal damage and subsequent (necrotic) cell death [10,11,65]. Other studies support mechanisms that focus more on disruption of autophagy or autophagic flux, leading to vesicle accumulation, defects in cellular energy balance due to autophagic failure, or defects in cargo recycling Evidence from cell culture [31,44] transgenic mice [44], and *Drosophila* [58] supports this hypothesis.

### 5.3. APOL1-Mediated Impairment of Mitochondrial Functions

Mitochondria are also critical for the survival of eukaryotic cells. Known as the powerhouses of the cell, mitochondria generate ATP through oxidative phosphorylation and are involved in regulating apoptosis, intracellular calcium homeostasis, and reactive oxygen species (ROS) detoxification (e.g., via superoxide dismutase). For *Trypanosoma,* it has been shown that APOL1 induces both lysosomal and mitochondrial membrane permeabilization [66]. Therefore, it is interesting that multiple studies have reported that APOL1 RRVs disrupt mitochondrial function also in mammalian cells. For example, expression of APOL1 RRVs in HEK293 cells or immortalized podocytes results in reduced mitochondrial membrane potential and due to reduced ATP levels, in decreased energy production [31,38]. This energy depletion led to increased permeability of mitochondrial membranes which compromises the ability of the Na^+^/K^+^-ATPase. Consecutively, this phenomenon was accompanied by an influx of chloride and water, which led to cell swelling [38]. A further study demonstrated that overexpression of APOL1 RRVs (lacking the SP) induced phosphorylation of stress-related kinase AMP-activated protein kinases, an indicator of cellular energy depletion and reduced cell viability. The same study also observed a correlation between decreased mitochondrial respiration and reduced intracellular potassium levels, suggesting that APOL1’s functions at mitochondrial membranes, are key contributors to RRVs associated cell injury [31]. In addition, there is evidence that APOL1 RRVs can oligomerize within the mitochondrial matrix, potentially forming pores or aggregates that interfere with normal mitochondrial function [41] and negatively influence mitochondrial fission [45]. Additionally, APOL1-mediated cell injury might to be connected to the putative role of the BSM that are present its PFD. BSMs have been linked to putative pro-apoptotic features by directly activating Bax/Bak proteins at the outer mitochondrial membrane to trigger apoptosis [67,68,69,70,71,72]. Hence, BSM and mitochondrial proapoptotic pathways may be a further mechanism beyond energy depletion explaining how mitochondrial dysfunction may mediate APOL1 RRV-dependent cell injury. In support of this, overexpression of APOL1 RRVs in which the BSM was deleted showed a reduced or even absent APOL1-linked cytotoxicity [3,73]. It is still under debate how the BSM contributes to APOL1-linked cell injury [72].

However, assuming that interaction with the outer mitochondrial membrane is required to mediate cytotoxicity, one would postulate a cytoplasm-facing membrane topology of APOL1 at intracellular membranes, since a luminal orientation of APOL1 would also result in luminal localization of the BSM.

### 5.4. APOL1 Mediated Disturbances of the ER Homeostasis

The ER represents the largest pool of intracellular membranes in eukaryotic cells and controls intracellular calcium (Ca^2+^) homeostasis (see above and Figure 3). Here, it serves as a major Ca^2+^ storage compartment that participates in both, Ca^2+^ release into the cytosol as well as its subsequent reuptake mediated by ATP-consuming pumps within the ER membrane (e.g., sarcoplasmic/endoplasmic reticulum calcium ATPase). Thus, the ER plays a key role in many Ca^2+^-linked cellular processes such as signal transduction (as second messenger), actin-dynamics, Ca^2+^-dependent activation of protein interactions or gene expressions. In addition to its role as Ca^2+^-dependent signaling hub, the ER performs a variety of essential and evolutionary conserved functions, including lipid and steroid production as well as protein synthesis, folding, and processing.

APOL1 is primarily localized to the ER, suggesting that it plays an important role in ER-associated functions [3,22,31]. This has led to the hypothesis that APOL1 RRVs may interfere with ER homeostasis and function. For example, Chun et al. (2019) proposed that wildtype APOL1 may act as a protective chaperone by transporting APOL1 RRV proteins away from the ER toward LDs (Figure 3), thereby reducing RRV-associated cytotoxicity. According to this model, damage arises when two RRV-coding alleles are present, due to the loss of the protective function provided by APOL1 wildtype [37,46]

In human cells, expression of APOL1 RRVs led to an increased expression of the chaperone HSPA, also known as GRP78 (78-kDa glucose-regulated protein) or BiP (binding Ig protein), accompanied by an elevated phosphorylation of the eukaryotic translation initiation factor 1 (eIF1) [74], which are indicators for an increased protein load in the ER with consecutive activation of the unfolded protein response (UPR). The UPR is an adaptive pathway that cells activate to cope with an increased protein load in the ER by three main mechanisms: first an increased folding capacity of the ER by enhanced expression of chaperon proteins, second accelerated degradation of misfolded proteins, and third reduced *de novo* protein biosynthesis.

In fly nephrocytes, where APOL1—like in human cells—predominantly localizes to the ER, high levels of APOL1 resulted in ER swelling, chaperone induction, and the expression of the ER reporter Xbp1-EGFP [57], once more indicators for the presence of ER stress linked to APOL1 overexpression. Activation of ER stress pathways, as indicated by the reporter construct, preceded cell death in nephrocytes. Support for a causal connection between high APOL1 RRV expression levels, ER stress, and cell injury comes from the observation that pharmacological inhibition of ER stress diminished APOL1-mediated cell death [57,74].

Intriguingly, ER stress is not inherently toxic but initially serves as a protective response to adverse conditions. When the ER becomes overloaded with misfolded proteins, the UPR is activated to restore ER homeostasis and protect the cell from further damage. However, if the stress persists and cannot be resolved, apoptosis may be triggered to remove damaged cells and maintain tissue integrity. In this way, high levels of APOL1 RRVs may shift the balance from protective UPR signaling towards proapoptotic UPR signaling and consecutively, increased cell death [50,57,74,75]. The pathogenic relevance of ER stress and maladaptive UPR signaling has been described in different compartments of the kidney and different entities of kidney disease. In acute kidney injury (AKI), ER stress was identified as a major mediator of progression from AKI to CKD as the presence of ER stress was linked to more pronounced damage and decreased recovery of kidney tubular epithelial cells, leading to increased kidney fibrosis [76]. In different AKI models, transgenic mice with ER stress activated by the stimulator of interferon genes (STING) showed accelerated inflammation of the kidney parenchyma that was associated with increased kidney fibrosis [77]. In patients with CKD due to hypertension and diabetes, a correlation between the expression levels of PERK (protein kinase R-like ER kinase) and ATF4 (activating transcription factor 4), two mediators of the UPR cascade, and the degree of kidney fibrosis was found. Furthermore, IRE1 (inositol-requiring enzyme 1) levels, also a UPR mediator, correlated with severity of CKD [77], indicating that a more pronounced activation of the UPR, shifting it from an adaptive to a deleterious UPR response can act as a driver of CKD progression. Also in podocytes, the cell type central to AMKD, ER stress was found in the context of diabetic nephropathy in mice, where an ER stress inhibitor was able to attenuate podocyte injury and reduce clinical as well histological parameters of glomerular kidney disease [78]. Similarly, in a mouse model for Alport syndrome, an inherited form of glomerular disease, ER stress was proposed as significant mediator of podocyte injury that was amenable to treatment with a chemical chaperone [79].

In summary, these findings indicate the ER stress is relevant for the development and the progression of kidney disease and that ER stress additional to APOL1 RRVs is not only a plausible pathogenic mechanism, but may also represent a promising target for therapy as already shown in a *Drosophila* model of AMKD [57]. Besides the protein overload in the ER, also the attenuation of *de novo* translation that results from increased phosphorylation of eIF2a and hampers the cells’ ability for protein replacement, may contribute to podocyte injury in AMKD. As podocytes are non-secretory cells, it remains an open question why this cell type appears to be particularly vulnerable to maladaptive ER stress, but likely this phenomenon is linked to the nature of podocytes and their specific vulnerability to disruption in pathways of cellular maintenance (see below).

### 5.5. APOL1 as a Regulator in Inflammation

*APOL* genes are part of the innate immune system (see above). Thus, it is tempting to speculate that APOL1 RRVs may trigger proinflammatory pathways and may damage kidney cells by increased inflammation, leading to cell death and tissue fibrosis. Indeed, it was shown that expression of APOL1-RRVs in podocytes *in vitro* leads to increased expression of proteins that are involved in the formation of inflammasomes. These protein complexes are part of the innate immune system and are responsible for promoting proinflammatory responses and death of infected host cells. APOL1-expressing podocytes showed not only increased expression of inflammatory mediators such as interleukin-1b and -18, but also induction of pyroptosis [80,81]. The relevance of this pathway in AMKD was supported by the observation that in transgenic APOL1 RVV mice, genetic inactivation of the inflammasome pathway reduced the kidney phenotype [82]. Interestingly, in these mice, there was a link between inflammasome activation and ER stress [77], indicating that uncontrolled proinflammatory signals in kidney cells may have a wider disruptive effect on other pathways that normally help restoring homeostasis. Hyperactivation of the innate immune system may thus be an additional mechanism of APOL1-RRV mediated cell injury that interestingly, could be linked to disruption of its ion channel function [81].

### 5.6. APOL1-Linked Cytotoxicity Due to Altered Lipid Biding

Several studies showed binding of APOL1 to lipids or lipid droplets [35,36,37,46]. Among the lipids that have been identified to bind to APOL1 were phosphatidic acid (PA) and cardiolipin (CL) as well as members of the phosphoinositides (PIPs), including PI4P, and cholesterol [35,36]. APOL1’s binding to lipids plays a central role in a pathomechanism proposed by Pays and colleagues, in which increased hydrophobicity of the APOL1 RRVs leads to podocyte injury (Figure 1c). The details of this concept are comprehensively summarized elsewhere [34,83]. In brief, here the APOL1 pathomechanism is predominantly based on hydrophobic interactions of intracellular APOL1 RRVs pools (according to splice variant vB3) with APOL3. Mechanistically, the APOL1 RRV-APOL3 interaction disrupts both, the role of APOL3 as an upstream regulator of Golgi PI(4)P kinase-B (PI4KB) activity as well as APOL3’s role in regulating mitochondrial membrane fusion and function. Moreover, dysfunction of APOL3’s tasks due to its binding to APOL1 RRVs is linked to several further downstream effects, including actomyosin reorganization, along with mitophagy and inhibition of apoptosis.

In addition to the cell-damaging role of intracellular APOL1 RRV pools, the increased expression of extracellular (probably paracrine) RRV pools also contributes to cellular damage. This occurs through the hydrophobic interaction of these pools with cholesterol-rich microdomains (lipid rafts), which in turn triggers a toxic Ca^2+^ influx and K^+^ efflux via TRPC6 and BK channels at the podocytes’ cell surface. In other words, in this concept APOL1 acts not as a pore or a channel but instead activates PM-localized cation channels. The increased binding of APOL1 RRVs to these cholesterol-rich PM pools may depend on a low HDL (high density lipoproteins)/high extracellular APOL1 ratios, which could occur both *in vitro*, in cell cultures and *in vivo* during interferon-mediated inflammation [34,83].

### 5.7. APOL1’s Role as a Membrane Pore or Ion Channel

There are a number of recent concepts, indicating that APOL1 can function as an ion pore or channel in both trypanosomes and human cell lines. The main idea here is that the APOL1 pathomechanism in human cells mirrors the one involved in trypanolysis. However, the specific ions primarily conducted by APOL1-RVVs associated pore or channel activity—whether chloride (Cl^−^), sodium (Na^+^), potassium (K^+^), or calcium (Ca^2+^)—remain unclear (excellently summarized in [84], see Figure 5). As mentioned, the structural organization of the PFD, alongside a MAD (Figure 1), resembles that of bacterial colicins [10,11]. Notably, this structural configuration is not unique to APOL1, but is also conserved across other members of the APOL family and indeed recent findings have demonstrated cation channel activity for APOL1–4, with variations in membrane insertion dynamics and ion conductance properties [85].

Thomson and Finkelstein observed that recombinant APOL1 induces pH-dependent, cation-selective conductance in planar lipid bilayers. The group also observed that insertion into the lipid membrane occurred under acidic conditions, followed by channel opening upon return to neutral pH values [26]. A further study found that when APOL1 interacts with lipids at a low pH and the pH is later neutralized, chloride permeability decreases, while potassium permeability increases [86]. These findings support a model in which APOL1 RRV-associated ion channels or pores are controlled in a pH-dependent manner. This is of high relevance as in mammalian systems, this implies that APOL1-induced ion pore activity initially requires insertion into likely Golgi or possibly endosomal membranes at acidic pH as well as subsequent trafficking to the cell surface and exposure to neutral pH to open cation-selective APOL1 channels (Figure 5b). In other words, such a passage through the endo-lysosomal system could be a precondition facilitating the potassium efflux that has been observed by different studies [24,28,38]. So far it is still unclear whether PM-localized APOL1 pools primarily facilitates Ca^2+^ influx [23] or K^+^ efflux with concurrent Na^+^ influx [28] (Figure 5f). In this context, it is worth noting that the identification of the primary ion flux in cellular systems is challenging, as disruption of ionic homeostasis may trigger compensatory responses [84].

Further support for APOL1 pore functionality also comes from the observation that Inaxaplin (also called VX-147) not only inhibits RRV-induced K^+^ efflux but also reverses APOL1-mediated cytotoxicity. Preliminary clinical data suggest that it may even mitigate the progression of APOL1-mediated kidney diseases in proteinuric patients [87,88]. These findings reinforce the hypothesis that the pathogenicity of APOL1 RRVs is primarily mediated through their pore-forming capacity and linked to its activity as an ion channel. Recently, the N264K variant was shown to suppress the cytotoxicity of the APOL1 RRVs—especially the G2 variant—via substitution of lysine at position p.264 with asparagine. This effect is primarily linked to the inhibition of potassium efflux at the cellular level and was found to reduce the risk of CKD in patients with APOL1 high risk genotype [89].

In light of the various, and on first view also contradictory mechanisms that have been suggested to explain APOL1’s cytotoxic cellular effects, a recent study proposed an elegant unifying model [28]. In this model, APOL1 is inserted into the PM, where it acts as a cation pore or channel that facilitates the transport of K^+^ and Na^+^ ions along with their natural concentration gradients. APOL1 RRVs result in an efflux of K^+^, combined with an influx of Na^+^ into the cell. This data fits to earlier observations in which imbalances of the cation homeostasis have been linked to APOL1 cytotoxicity (see above). The disturbances of the Na^+^ and K^+^ concentrations cause depolarization of the PM, which in turn activates a G protein-coupled receptor (GPCR) that initiates a phospholipase C (PLC) dependent production of the second messenger IP3 (inositol 1,4,5-trisphosphate). Binding of IP3 to its receptor (IP3R) at the ER leads to the release of Ca^2+^ from ER stores into the cytoplasm of cells expressing APOL1 RRVs. This Ca^2+^ release—which is further amplified by an activation of the ryanodine receptor (RyR)—leads to a reduced mitochondrial function. This aspect combines recent observation that an increased intracellular Ca^2+^ concentration [23] and mitochondrial dysfunction along energy depletion are key players in APOL1-linked cytotoxicity. The Ca^2+^-dependent mitochondrial dysfunction, in turn, which causes a decline in intracellular ATP levels triggers the energy-sensing kinases AMPK (5’ adenosine monophosphate-activated protein kinase) and CaMKKβ (Ca^2+^/calmodulin-dependent protein kinase kinase β) and also promotes an increased autophagy as well as inhibits the mTOR (mechanistic target of rapamycin) kinase. This reduces global protein synthesis. Additionally, AMPK activation enhances phosphorylation and therefore activation of various stress kinases such as p38 and JNK (c-Jun-N-terminal kinase). Depolarization of the PM, along with ATP depletion, also impairs the uptake of amino acids via amino acid/Na^+^ co-transporters [28]. Thus, Olabisi and colleagues proposed a model that connects most of the previously published mechanisms to explain how APOL1 RRVs cause cellular damage.

Open question: Nevertheless, central questions remain unresolved, particularly concerning how increased ion transport or pore activity connects mechanistically with RRV-associated disruptions of endocytosis, autophagy, pyroptotic cell death, and mitochondrial dysfunction. Furthermore, the exact mechanisms by which APOL1 is transported to the cell surface are still unclear and how (or in which cellular compartment) the pH shift required for pore activation are achieved. Moreover, it remains to be shown if these surface-localized pools are the sole source of APOL1’s cytotoxic effects, or if APOL1 affects also other membranes or cellular organelles. Moreover, do APOL1 pores arise from dimers or higher order multimers, by a pH-independent mechanism? And finally, what molecular events trigger or stabilize this oligomerization and how the C-terminal region that carries the RRVs involved in these processes?

## 6. APOL1-Associated Cytotoxicity in Podocytes and the Role of Second Hits

AMKDs are a group of kidney diseases that not only affect humans that are positive for two APOL1 RRVs. However, in the presence of RRVs these diseases show accelerated and more aggressive clinical courses. AMKDs share common clinical features, particularly the presence of proteinuria and progressive CKD and include glomerular diseases such as HIV-associated nephropathy (HIVAN), COVID-19-associated nephropathy (COVAN), focal segmental glomerulosclerosis (FSGS), lupus nephritis, hypertension-associated nephropathy, and membranous nephropathy [17,55]. Proteinuria reflects damages of the renal filtration barrier, suggesting that podocytes are key target cells of APOL1-associated cytotoxicity. The clinical manifestations and differences among AMKDs have been extensively reviewed elsewhere and will therefore not be addressed in detail here [18].

In summary, these differences strongly suggest that, while the homozygous presence of two APOL1 RRV confers an increased susceptibility to different types of kidney disease, the actual onset and progression of AMKD are significantly influenced by the underlying primary disease cause, environmental, and other factors (see above). Therefore, it has been hypothesized that so-called "second hits" are necessary for the earlier onset and progression of AMKDs.

So far, two genetic modifiers have been described. Zhang et al. identified the *UBD* (ubiquitin D) locus as a genetic modifier of AMKDs and observed that people with two RRVs showed an increased susceptibility for AMKD, when they had increased African ancestry at the *UBD* locus, which correlated with lower expression levels of *UBD* [90]. Further, it was found that truncated APOL3 increase the risk for CKD in the presence of APOL1 RRV [91,92]

Nevertheless, to a large extent, the nature of these secondary factors remains unknown and, beyond genetic modifiers that are yet to be identified, environmental influences may play a substantial role. Given the heterogeneity of AMKDs, one would postulate that beyond common genetic or environmental factors, there will also be disease-specific, and treatment associated factors, such as e.g., the control of disease activity in lupus nephritis or in HIVAN.

In contrast to the clinical situation, *in vitro* and *in vivo* models of APOL1-RRV demonstrate a robust phenotype of cellular injury without the need for additional factors. Therefore, it is valuable to discuss the second-hit hypothesis not only from a clinical perspective but also from a cell biological point of view, which clearly distinguishes between the genetic level (or genetic predisposition) and the functional level*,* which is based on protein expression and subsequent cell biological consequences. This leads to an interesting and perhaps neglected aspect: Since in individual cells, the expression of APOL1 RRVs causes cell death without a second hit, a plausible explanation could be that APOL1 RRV expression and its subsequent various pathological effects are themselves the second hits, while other factors, such as inflammatory, immunologic, or hypoxic stimuli, may be the first hits as they drive the necessary upregulation of APOL1 at the protein levels *in vivo* [2,21,93,94]. As mentioned, almost all *in vivo* and *in vitro* models show a cytotoxic effect upon overexpression, whereas in AMKDs, APOL1 RRVs are more likely to be responsible for faster or more severe disease progression. How do these findings fit together?

One possible explanation could be that in AMKDs, at the cellular level, pre-damaged or exhausted cells are present, which are further injured by the expression of APOL1 RRVs, which in turn results in increased loss of cell viability or even (programmed) cell death. Such pre-damaged cells could result from viral infections, during which a significant portion of energy resources is redirected to support viral proteins and nucleic acids. Additionally, ER stress or immunological stimuli could also cause cellular exhaustion, where the forced additional expression of APOL1 RRVs induces further harm, overwhelming the cells’ resilience potential.

Another possible explanation (perhaps in combination with the above-mentioned one) that aligns well with the predominant kidney phenotype, is that podocytes are particularly susceptible to injury mechanisms compared to other cell types (e.g., hepatocytes, endotheial cells) that have been associated with APOL1 RRVs [50,52,95,96,97,98]. Podocytes are highly specialized, terminally differentiated cells that, together with glomerular endothelial cells and the glomerular basement membrane, constitute the renal filtration barrier. Indeed, from a cell-biological perspective, podocytes are distinct from other renal cells due to their postmitotic nature, their complex architecture and their requirement for sustained cellular homeostasis [99,100,101]. Their complex morphology includes interdigitating, actin-rich foot processes that are connected by the slit diaphragm (SD), a unique cell–cell junction critical for maintaining filtration selectivity. In particular, the formation and maintenance of the foot processes and the SD depend on the precise biosynthesis and trafficking of specific transmembrane proteins, which must be correctly processed in the ER and targeted to cholesterol-rich membrane microdomains at the SD. This places a high demand on ER function, making podocytes particularly vulnerable to disturbances in protein-folding homeostasis and ER stress. Moreover, an elaborate endolysosomal and autophagic system supports the recycling and localization of SD components [102].

Podocyte injury initially leads to foot process effacement, accompanied by reduced attachment to the basement leading an increased glomerular leakage [103]. On a molecular level this is linked to dysfunctional re-arrangements of the actin-cytoskeleton. More severe damage causes detachment or apoptosis of podocytes, culminating in irreversible glomerular sclerosis. These specialized features, while essential for podocyte function, may also render them particularly susceptible to the cytotoxic effects of APOL1 RRVs.

Open questions: While this is an intriguing concept, several open questions remain open including why the various diseases—especially those involving podocytopathies—activate signaling pathways that subsequently lead to APOL1 upregulation. In a way, this shifts the question toward identifying the primary molecular and cellular causes of these podocyte diseases. In case of viral infections, the initial trigger is clear. However, in autoimmune or autoinflammatory diseases, or in the various forms of FSGS, the situation is by far more complex and requires a deeper understanding of which disease-signaling-pathways are broadly associated with AMKD and AMKD-associated podocytopathies.

Furthermore, it is necessary to investigate in more detail how the exact interplay between “pre-damaging” factors and the injury mediated by APOL1 RRVs occurs. The question of why RRVs remain dormant for many years, leading to a relatively late onset of kidney disease, is only partially understood at the cellular level. It is important to identify where the threshold for podocyte injury lies, and which potentially modifiable factors might act as additional elements to delay or prevent the harmful effects of APOL1 RRVs. A precise understanding of these details would also provide a crucial basis for developing therapeutic strategies against AMKDs.

## 7. Concluding Remarks

Taken together, APOL1 RRVs mediate a plethora of intracellular events that molecularly drive cell injury. In the review, we identified numerous cellular biological aspects regarding APOL1 that have been poorly addressed so far, and for which it may be important to conduct more in-depth investigations. All questions lead to the fundamental problem, why these effects are particularly causing kidney failure and, on a cellular level, podocyte injury. Is this phenomenon due to a particular feature of APOL1 RRVs that only comes into play in podocytes? Is it due to a particular combination of splice and risk variants? A unique interaction partner? A specific subcellular milieu? A distinct pattern of intracellular transport or protein degradation? Or is it simply that exhausted podocytes, due to their postmitotic nature and complex cellular morphology, are particularly vulnerable to damaging properties of APOL1 RRVs, while other cell types have more compensatory capacity or resilience mechanisms. Finally, current studies using patient material are primarily focused on histological analyses and transcriptomic profiling. These studies should be expanded to include single-cell approaches and high-resolution imaging of human specimens derived from AMKD patients to gain deeper cell biological insights.

## Figures and Tables

**Figure 1 cells-14-00960-f001:**
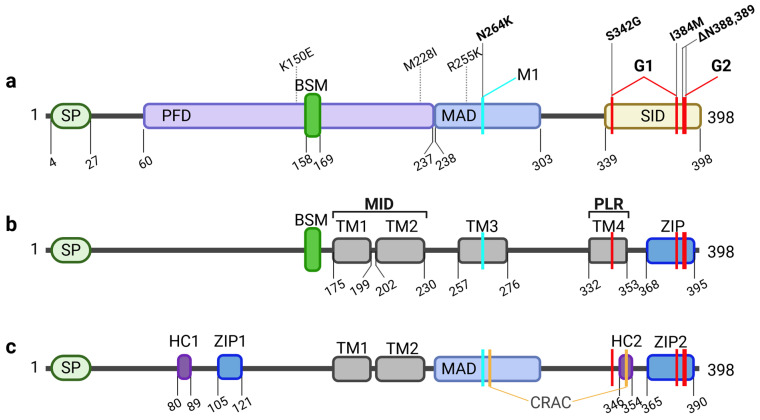
Structural features of APOL1. (**a**) The regions described first relate to functional domains of APOL1. Besides the signal peptide (SP) in the N-terminus, important for secretion, the sequence contains a pore forming domain (PFD), which includes the Bcl-2 homology 3 (BH3) domain-like sequence motif (BSM) associated with putative anti-, or pro-apoptotic features. The PFD and neighboring membrane-addressing domain (MAD) enable the membrane localization and pore activity. C-terminally localized is the SRA-interacting domain (SID). SID is important for SRA binding. Naturally occurring variants of the African haplotype (K150E, M228I, R255K) are located in the PFD and MAD, while renal risk variant G1 (S342G, I384M) and G2 (ΔN388,389) are localized in the SID. In turn, modifier 1 (M1; N264K) which prevents G2-mediated toxicity is localized in the MAD. (**b**) A more topological classification of APOL1 regions focusses on up to four putative transmembrane helices (TM1-4) in addition regions to the SP, which determines the topology of the APOL1 splice variants. Juxtaposed to the BSM, TM1 and TM2 depict the membrane insertion domain (MID) followed by TM3 harboring the N264K variant and TM4 also defined as the pore lining region (PLR) enabling pH-sensitivity and selectivity of the pore. Here, one of the G1 variants (S342G) is located, while the other (I384M) appears together with the deletions causative for G2 in a leucine zipper (ZIP) domain in the C-terminus, probably involved in pore formation via APOL1 multimerization. (**c**) A mixed functional-topological model by Pays and colleagues includes a combination of a hydrophobic cluster (HC) paired with a close-by ZIP domain located in the N-terminal part as well as in the C-term. APOL1 also contains two putative cholesterol recognition amino acid consensus (CRAC) sites in the MAD and the HC2. Following this model, membrane insertion is enabled by TM1, TM2 and the MAD.

**Figure 2 cells-14-00960-f002:**
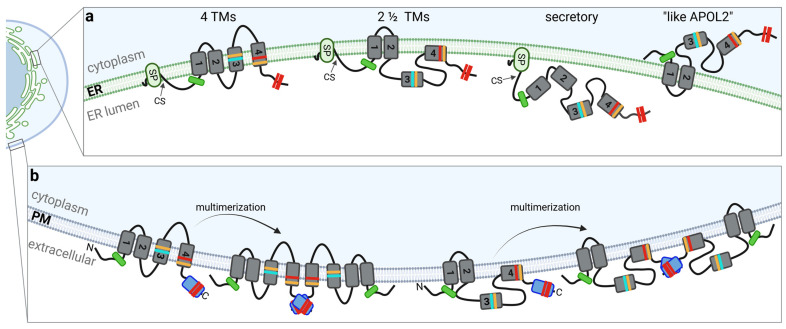
Topologies of APOL1 at membranes. The colors used in this figure for characteristic APOL1 features (domains, variants, etc.) corresponds to that of Figure 1. (**a**) In the 4-TM concept, four transmembrane regions or helices (TMR1-4) span the ER membrane. In the two and a half-spanning loop model (2 ½ TM) two TMRs are present in the PFD, followed by a region between MAD and SID that “dips” into the membrane. Secreted or luminal pools of APOL1 contain a signal peptide (which is cleaved inside the ER, cs) and are only associated (but not inserted) into intracellular membranes. In these three models, all sequence motifs associated with APOL1-linked cytotoxicity (including the BSM, the RRVs, G1 and G2, the amino acids that determining the *African* haplotype, as well as the two cholesterol-binding sites; see Figure 1) facing the luminal side of intracellular membranes. The fourth concept refers to a cytoplasmic-faced APOL1 topology with only two TMRs (or membrane associated regions), resulting in an orientation in which both the N- and C-terminus as well as all mentioned cytotoxicity-associated sequence motifs facing the cytoplasm (like APOL2). (**b**) In case that pore formation requires dimerization or even multimerization both the 4 TM and 2 ½ TM concepts facilitate a “tail-to-tail” multimerization via a C-terminal ZIP domain.

**Figure 3 cells-14-00960-f003:**
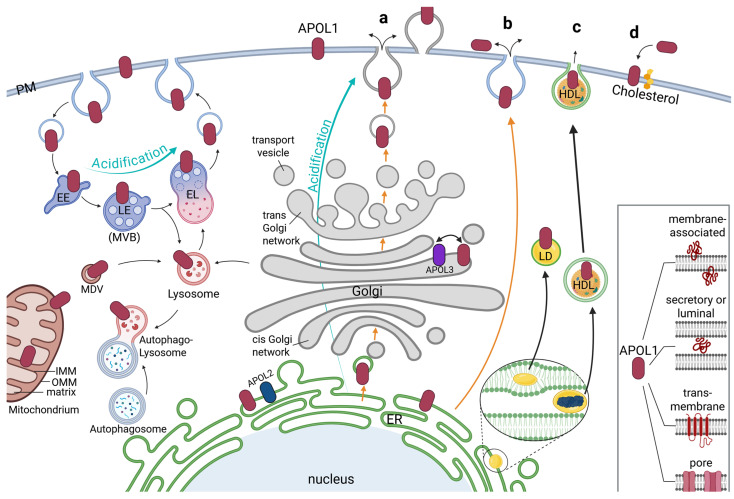
Localization, trafficking and secretion of APOL1. The main pools of APOL1 are detected at the endoplasmic reticulum (ER) with further portions reported at the plasma membrane (PM), early/late endosome (EE/LE, also called multivesicular body, MVB), lysosome and endolysosome (EL), mitochondria and Golgi apparatus as well as at Lipid droplets (LD) and high-density lipoprotein (HDL) particles. Recycling and degradation could be mediated via the endolysosomal-autophagic system. Acidification of APOL1 may take place in the endolysosomal pathway or the conventional secretion via the Golgi apparatus (turquoise arrows). Mitochondrial APOL1 could integrate both into the outer or inner mitochondrial membrane (OMM, IMM) and transported via mitochondria-derived vesicles (MDV). Possible pathways for APOL1 secretion (orange arrows) could be conventional secretion (**a**), or unconventional secretion (**b**). In addition, transport via LD or HDL particles is possible (thick black arrows). Both arise within the ER, which leads to a lipid mono- or bilayer perhaps containing ER-located APOL1 proteins encapsulating them when released (**c**). A reported association of APOL1 with cholesterol, which is produced in the ER may also be part of the trafficking and release of APOL1 (**d**). Inset: Independent of its localization, APOL1 can appear as a membrane-associated protein (in luminal or cytoplasmic orientations), as secretory or luminal protein, as an integral transmembrane protein, or even as pore-forming protein.

**Figure 4 cells-14-00960-f004:**
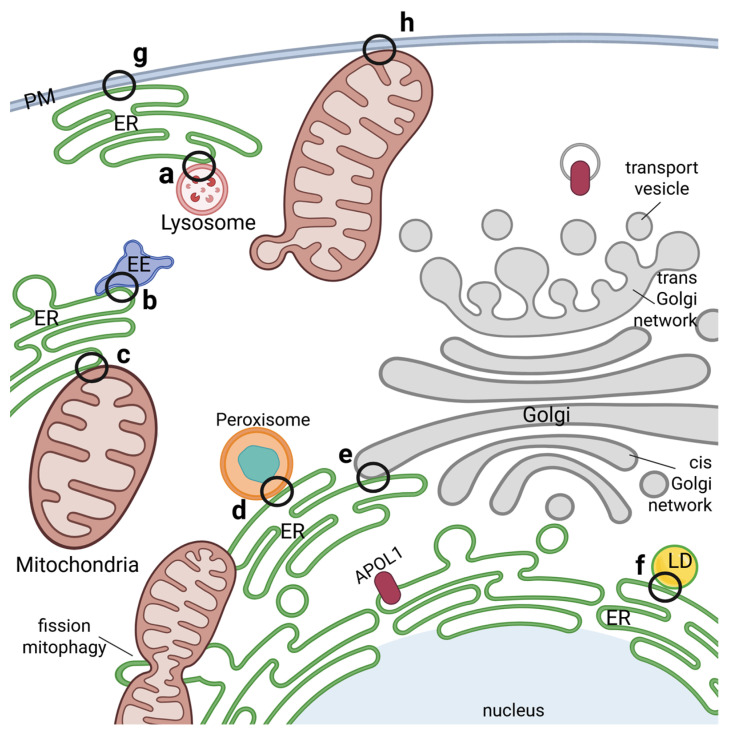
Membrane contact sites between organelles as potential APOL1 transfer hubs. Membrane connections between different organelles in close proximity are mediated via membrane contact sites (MCSs) and influence signaling, ion homeostasis, metabolism, stress responses as well as function, division and regulation of organelles. Such MCSs have been identified between ER and lysosomes (**a**), endosomes (**b**), mitochondria (**c**), peroxisomes (**d**), the Golgi network (**e**) or lipid droplets (**f**). MCSs can also be formed between the PM and ER (**g**) as well as PM and mitochondria (**h**). APOL1 localizes in/at nearly all the compartments mentioned and could either play a direct role in these MCSs or be transferred there from the cytoplasm or one compartment into another. Such a way of transfer has been described e.g., for mitochondrial protein import. Interestingly, APOL1 was already reported to play a role in mitochondrial fission and mitophagy.

**Figure 5 cells-14-00960-f005:**
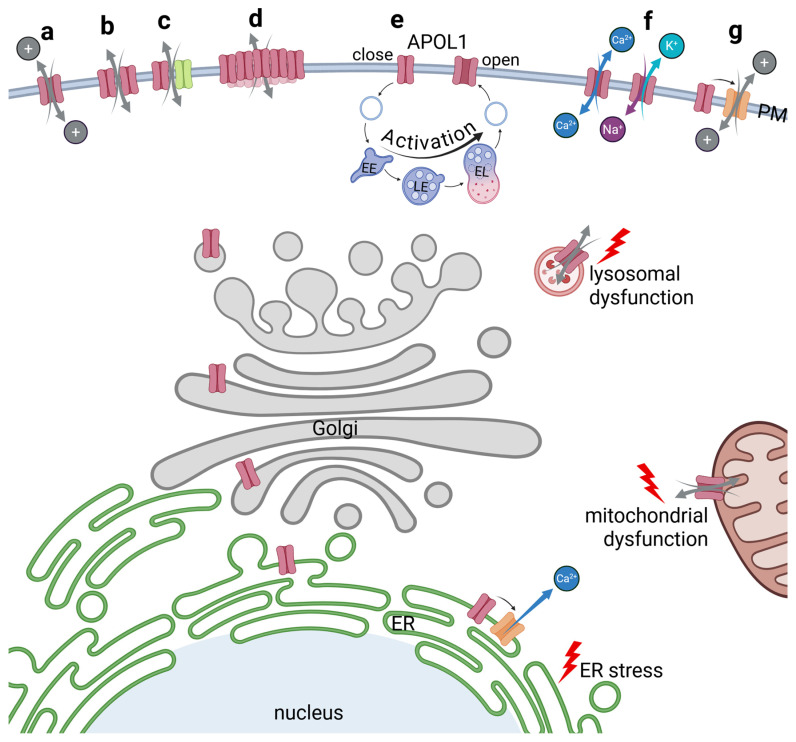
The function of APOL1 as an ion pore. APOL1 pathomechanism includes ion pore functions of the APOL1 renal risk variants. The first possibility could be that the TM-helices of APOL1 themselves form a pore within the PM (**a**). Pores could also be built by homodimerization of APOL1 subunits (**b**), or by heterodimers composed of APOL1 and a second protein (**c**). As a APOL1clustering at the surface has been reported, numerous APOL1 proteins could also be subunits in a multimerized assembly into a megapore (**d**). Some studies suggest a pH-dependent formation of the pore, which would require involvement of the endolysosomal pathway to acidify and open the APOL1 pore (**e**). Several experiments hint towards a pore selectivity for cations which could for example facilitate Ca^2+^ flux or K^+^ efflux combined with Na^+^ influx (**f**). Indirect effects due to activation of nearby ion channels could also be possible (**g**). All mechanisms bear the possibility that APOL1 not only serves as pore at the PM, but also at other membrane-containing compartments leading to lysosomal or mitochondrial dysfunction or ER stress.

## Data Availability

Not applicable.

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
