# Peer review of "A Cell Biologist’s View on APOL1: What We Know and What We Still Need to Address"

_cells, 2025, doi:10.3390/cells14130960_

Round 1
Reviewer 1 Report
Comments and Suggestions for Authors
see uploaded

Very written overall but some of the more complex sentences are not as a native English speaker would write and are difficult to read. I'd be happy to point them out, but the reviewer's instructions said to leave that to the editors.
Author Response
Point-to-point-response (reviewer 1)
Review of Höffken et al “A cell biologist’s view on APOL1: what we know and what we still need to address”. Höffken et al provide an extremely thorough and enjoyable review of the cell biology of APOL1 and how the renal risk variants might cause APOL1-mediated kidney disease. They cover the majority of relevant research in an unbiased fashion and clearly point out our current gaps in understanding. The discussions on APOL1 topology, form of insertion and/or binding to membranes and intracellular localization with respect to function are particularly well written, novel and thought-provoking and, in my opinion, long overdue.
Response: We are very pleased with the highly positive evaluation of our cell biological review on APOL1.
There are only a couple of major points to clear up. Most importantly, the use of cis and trans to describe APOL1 variant and family member topology. This terminology is not commonly used in the renal field and will likely confuse most readers, particularly as I think the authors themselves have got them the wrong way round when referring to intracellular pools (albeit correct when referring to extracellular pools of APOL1). I believe cis should refer to proteins translated on the same side of the membrane as the ribosomes (i.e. cytoplasmic side on free ribosomes), and cis APOL1 (such as vB3) will remain cytoplasmic whether attached to the cytoplasmic surface of the ER or the inner plasma membrane. Trans should refer to APOL1 variants with a signal sequence that are co-translationally translocated into the ER lumen on rough ER ribosomes (topologically equivalent to the extracellular space, whether fully secreted or inserted into the plasma membrane with its N- and C-termini facing out). Thus, it would be better if Fig2a was redrawn the other way up so that ER lumen is on the same side ( i.e. upper) as Fig2b’s extracellular side and the APOL1 proteins are drawn in the same orientation in both panels (thus the mixed up trans/cis labels in 2a would become correct). However, given that cis and trans ER is more commonly used to refer to nuclear membrane ER versus Golgi-facing ER, it may be better to omit the use of cis vs trans altogether and just call it cytoplasmic vs luminal/extracellular to avoid confusion.
Response: We agree with the reviewer. The use of “cis” and “trans” orientation unnecessarily complicates the topology issue and does not contribute to better readability. Therefore, we have revised all relevant passages and removed the terms cis and trans (including figures and figure legends). Instead, we now consistently use the terms luminal or extracellular when the N- and C-termini of APOL1 are found in the ER lumen and later on the extracellular side of the cell surface. A cytoplasmic orientation is used when the N- and C-termini point into the cytosol. We hope this makes the topology-related aspects clearer and easier for the reader to understand.
(see yellow marked sentences: page ¾ (legend), 104-124; page 5, 178-182; page 6, 229-233; page 12 470-473)
The authors should include the possibility that APOL1 (with signal peptides) is inserted into the membrane during transit through the secretory pathway (as hypothesized by the Raper lab, e.g. Giovinazzo et al ref 25) because the secretory pathway becomes more and more acidic from the cis thru trans Golgi thru secretory vesicles and is in fact comparably acidic to the endosomal pathway (Paroutis, Touret & Grinstein (2004) Physiology 19 p207). Thus, transit through the endosomal system is not necessarily a requirement for membrane insertion and cation pore formation. As mentioned in Fig3, it is unclear whether APOL1 would be peripherally attached to the inner ER membrane, or have at least TMs 1&2 inserted (if not 2 ½ or 4 TMs), but the Raper lab’s in vitro data strongly suggest the ionic pore formation (TM4) event will not occur until low pH is encountered. This is backed up in stable APOL1 cell lines by the use of ammonium chloride (Giovinazzo).
Response: We thank the reviewer for this very important and often overlooked point. Indeed, not only passage through the endo-lysosomal pathway but also the gradual acidification during the secretory pathway is linked to stepwise acidification, influencing APOL1’s the insertion into cellular membranes. We have included this aspect in the text of manuscript and in the figures and also cited the referenced literature from the Grinstein Laboratory.
Minor points:
Comment 1) Line 29: add “protection from” between on one hand and HAT otherwise it sounds like APOL1 causes HAT
Response 1) We thank the reviewer for this comment and have made the corresponding changes.
Comment 2) Line 49 isn’t clear what the alignment is of. Should read “genomic sequences of APOL1 thru APOL6 were aligned with each other to look for phylogenetically….”
Response2) We thank the reviewer for this comment and have made the corresponding changes.
Comment 3) 67: on chromosomes 6 and 12 in mice and humans, respectively [either remove “respectively” or remove “and chromosome”]
Response 3) We thank the reviewer for this comment and removed the “respectively”.
Comment 4) Line 81/82 expression, not expressed
Response 4) We thank the reviewer for this comment and have made the corresponding changes.
Comment 5) Line 90: add “so-called” in front of pore-forming domain since the Raper lab has shown
that the actual APOL1 pore is in TM4/SRA-ID, not in the colicin homology domain
Response 5) We agree an inserted “so called” to emphasize this aspect in the manuscript text.
Comment 6) Line 95: replace “in parasites resistant to APOL1, such as certain Trypanosoma subspecies” with “In Trypanosoma brucei rhodesiense” since Tbr is the only subspecies with SRA.
Response 4) We thank the reviewer for this comment and have made the corresponding changes.
Comment 7) 108: SID is important for SRA binding (“immunological function against Trypanosoma” is confusing since SRA prevents immunity)
Response 7) We thank the reviewer for this comment and changed the text accordingly.
Comment 8) 125: I would remove “is most likely mediated” from secretion by an N-terminal signal peptide as these very authors have demonstrated experimentally the functionality of the signal peptides for all isoforms.
Response 8) We removed “most likely”, as suggested.
Comment 9) 133: Tbr and Tbg separately evolved resistance to APOL1 [not necessary to mention but Tbg’s resistance mechanisms include a mutation in TgsGP that decreases membrane fluidity to prevent APOL1 insertion, increased cysteine protease activity in lysosomes to digest APOL1 and mutation of the TLF-1 receptor HpHbR that decreases TLF-1/APOL1 uptake]
Comment 10) 137: G2 enables lysis of Tbr but not Tbg by preventing Tbr’s SRA binding (thus selfexplanatory that it doesn’t affect Tbg)
Response to comments 9) and 10): We thank the revier for this comment and changed the passage into
“Two subspecies of Trypanosoma brucei — T. b. rhodesiense and T. b. gambiense — have evolved resistance to APOL1 by producing different forms of their main surface protein, called VSG (Variant Surface Glycoprotein). These forms are known as SRA in T. b. rhodesiense and TgsGP (T. gambiense-specific Glycoprotein) in T. b. gambiense. SRA directly binds to the C-terminal part of APOL1, blocking its function, while TgsGP decreases membrane fluidity of the parasite, which prevents APOL1 insertion. In response, two APOL1 sequence variants called G1 (S342G/I384M) and G2 (Δ388N, Δ389Y) emerged in humans of Sub-Saharan African ancestry approximately 10,000 to 20,000 years ago. These changes within the C-terminal region avoid SRA binding and restore resistance to T. b. rhodesiense.”
(page 4, 133-142)
Comment 11) 153: evolution, not evolvement
Response 11): We have corrected this typo.
Comment 12) 164: APOL1’s insertion into ER lumen and its subsequent transport to the plasma
membrane and/or release into the extracellular medium
Response 12) We thank the reviewer for this comment and changed the sentence/text accordingly.
Comment 13) 174: luminal is trans not cis
Response 13) We agree, the reviewer's point is correct (see above major revision): the distinction between “cis” and “trans” topologies unnecessarily complicates the APOL1 orientation aspect (see above). Therefore, we revised this part of the manuscript and consistently removed the “cis” and “trans” words. We now simply refer to a “luminal or extracellular orientation” and an orientation facing the cytoplasmic side (cytosol)"
Comment 14) 187: “dips into the membrane of podocytes”
Response 14) We thank the reviewer for this comment and included “of podocytes”
Comment 15) 190: 4th TMR becoming a pore lining region in planar lipid bilayers [because mentioning these very different membrane systems may help explain the differences in topology since naturally occurring co-translationally inserted APOL1 may not be in the same orientation as recombinant APOL1 added to bilayers after folding even if both encounter acid]
Response 15) We thank the reviewer for this comment and included “in planar bilayers”
Comment 16) 197: replace “combined to” with “followed by”
Response 16) We replaced “combined to” with “followed by”, as suggested.
Comment 17) 202: trans-orientation, not cis [or just delete]
Response 17) We deleted this (see also response to comment 13)
Comment 18) 205: cis not trans [or just delete]
Response 18) see response to comment 13)
Comment 19) 206: replace “are faced to” with “facing”
Response 19) We replaced “are faced to” by “facing”, as suggested.
Comment 20) 207: multimerization both the 4TM and 2 ½ TM…
Response 20) We thank the reviewer for this comment and changed the sentence/text accordingly.
Comment 21) 208: Add sentence saying that the domain color scheme is the same as in Figure 1.
Response 21) We thank the reviewer for this comment and added the following sentence: “The colors used in this figure for characteristics of APOL1 features (domains, variants, etc.) corresponds to that of Figure 1.”
Comment 22) Figure 2: #3 in TM3 is obscured by the color scheme, can it be brought to the front?
Response 22) We revised the Fig. 2, as suggested.
Comment 23) Fig 2a – draw other way up as described above
Response 23) We revised the Fig. 2, as suggested.
Comment 24) 219 & 221: cis not trans
Response 24) see response to comment 13)
Comment 25) 230: worth pointing out that all these putative binding partners are cytoplasmic (although a portion of NM2A is also in the inner mitochondrial membrane), as this suggests that the major secretory isoform does not bind any of these in the secretory pathway. [APOL1 is known to be very sticky, so can bind many things in vitro; we have unpublished data of APOL1 immunoprecipitates where the Coomassie gel looks like a lysate with various very clean antibodies!].
Response 25) We addressed this point and added the following sentence: “Strikingly, APOL3, NM2A, and PHB2 are cytoplasmic proteins, while VAMP8 is a membrane protein with relevant domains facing the cytoplasm, indicating that the SP-containing APOL1 isoforms do not bind to any of them.”
(Page 6,238-241)
[In addition, we thank the reviewer for this unpublished data/information]
Comment 26) 239: it is in fact known that multiple APOL isoforms are expressed in the same cell type (albeit all in vitro cell lines). APOL1.vA is by far the most abundant, followed by vC then vB3 as assessed by PCR in podocytes, endothelial cells, RPTECs and others (Nichols ref 23, Cheatham ref 22; and Scales ref 24 Fig S20, which quantitatively shows vA is 30x more abundant than vC and 70x vB3). Nonetheless, it is indeed unknown what the relative levels are in different tissues in vivo.
Response 26) We agree and revised the corresponding “open questions passage” as follows: We removed the sentences “It is also unknown whether the different splice variants are simultaneously expressed in a single cell, or whether certain splice variants are exclusive to specific cell types. If simultaneous expressions occur, it would also be important to determine the relative abundance of each splice variant within the cell.”
We modified/revised the “open question passage” as follows:
“Open questions: So far, it remains unclear what factors control or trigger the splicing of the APOL1 gene in vivo. Furthermore, it still needs to be determined whether ER luminal or cytoplasmic orientations are restricted to the ER and plasma membranes, or if they are also present on membranes of other cellular compartments. It also remains to be shown what the relative levels are in different tissues in vivo. Even if the main APOL1-associated cytotoxicity originates from the SP-containing APOL1 vA splice variants, it should be clarified to what extent the other variants contribute to or modify cell-damaging effects. Finally, it still needs to be shown how high the relative concentrations are in different tissues in vivo.”
(page 6/7, 250-260)
Comment 27) 243: orientations are restricted to the ER and plasma membranes, or…
Response 27) We included “and plasma membrane”, as suggested.
Comment 28) 267 (fig 3): it’s unclear from the figure what the difference between “a” (conventional) and “b” (unconventional) secretion is
Comment 29) Fig 3: consider also adding the possibility that APOL1 is shed in microvesicles (like “a”
except vesicle buds outwards instead of inwards).
Response to comment 28 and 29) We revised the Fig. 3 as suggested.
Comment 30) Fig 3: consider also the possibility that APOL1 could be secreted onto HDL particles by ABCA1 transporters like APOA1 and APOE
Response 30) This is an extremely interesting aspect that requires further research. However, taking these pathways into account would make the figure even more complex than it already is. Therefore, we decided not to include this special aspect in the figure.
Comment 31) Fig 3: draw an “Acidification arrow” through the Golgi and secretory vesicles like you have for the endosomal pathway
Response 31) We revised the Fig. 3 as suggested.
Comment 32) 297: APOL1 pools are found at the ER and Golgi membranes [approx. 18% in Golgi by EM, Scales ref 24]
Response 32) We included “and Golgi”, as suggested by the reviewer.
Comment 33) Fig 4 – need to write a,b,c etc bigger (took me longer to find the letters than the organelles themselves). Other text in this figure should also be a bit bigger
Response 33) We revised the Fig. 4 as suggested.
Comment 34) 322: plasma membrane APOL1 was also shown in Ref 24 (by EM and FACS)
Response 34) We now included Ref 24.
Comment 35) 333: Previous not pervious
Response 35) We now have corrected this typo.
Comment 36) 354: secreted has no “a” in it
Response 36) We now have corrected this typo.
Comment 37) 359: consider mentioning the possibility that the abundant ER APOL1 simply reflects slower secretion (i.e. packaging into transport vesicles) than translation? For example, it could be that only APOL1 with enough TMs inserted into the membrane folds correctly to pass ER quality control for export, while peripherally associated (“like APOL2”) APOL1 does not.
Response 37) We are grateful for this excellent comment. Indeed, an imbalance between translation at the ER, membrane insertion of APOL1 into the ER membranes and APOL1’s subsequent export/secretion towards the cell surface could be a main factor why scientist observed such high amounts of APOL1 at ER membranes. We included this important point now in the revised manuscript text. The revised section now reads as follows:
Open questions: APOL is found in various intracellular pools, …
A key question in this context will be to address whether an imbalance between APOL1 translation at the ER, its insertion into ER membranes, and its subsequent export to the cell surface might be responsible for the high intracellular APOL1 pools, particularly at ER membranes. There is also a gap in our understanding of the cell biological details of how APOL1….
(page 10, 379-382)
Comment 38) 404: may be worth mentioning that the lack of rescue of RRVs by wild type APOL1 was performed with non-African wild type vs African RRVs (African G0 might have rescued?)
Response 38) Indeed, the role of the haplotypes is complex and should be considered. We addressed this point by including "(non-African)" in brackets in the sentence: "The latter seems more plausible, as some evidence also indicates that high levels of (non-African) APOL1 wildtype do not rescue cells from RRV-induced toxicity."
Comment 39) 450: cis and trans wrong way round
Response 39) see response to comment 13)
Comment 40) 512: chaperone
Response 40) We now have corrected this typo.
Comment 41) 535: link, not cross-link
Response 41 We now have corrected this typo and replaced “cross-link” by “link”.
Comment 42) 548: “in this special issue” implies the references 36,85 are referring to other articles in this issue, when in fact they are a year or more old
Response 42) We removed “in this special issue”.
Comment 43) 576: only APOL1-4 were directly shown to act as cation channels (APOL5 couldn’t be made in sufficient amounts and APOL6 lysed the artificial membrane)
Response 43) We thank the reviewer for this important point and changed “APOL1-6” in “APOL1-4”.
Comment 44) 586: requires insertion into likely Golgi or possibly endosomal membranes
Response 44) We changed this sentence as suggested.
Comment 45) 590 so far it
Response 45) We now have corrected this typo.
Comment 46) 651: It’s very unlikely that the ER is the source of APOL1 cytotoxicity since Giovinazzo’s elegant RUSH-APOL1 system (ref 25) and Kruzel-Davila’s (ref 31) and Gupta’s (ref 32) BFA experiments prove that APOL1 must at least exit the ER to be toxic (and likely also reach the plasma membrane, although technically we can’t exclude effects in the Golgi). Additionally, APOL1 in the ER has not yet met an acidic compartment to form a pore.
Response 46) : We agree that it is unlikely that the endoplasmic reticulum (ER) is the primary source of APOL1 cytotoxicity. However, it is likely that the ER is indirectly involved in several mechanisms associated with APOL1-induced cytotoxicity. The ER regulates how much APOL1 is synthesized and subsequently transported to cellular membranes; it serves as the central intracellular Ca²⁺ store; it maintains contact with nearly all other organelles; and it plays a general role in proteostasis, including protein folding, processing, and quality control.
Top adress the reviewer’s concerns we changed this sentence into:
“Moreover, it remains to be shown if these surface-localized pools are the sole source of APOL1’s cytotoxic effects, or if APOL1 affects also other membranes or cellular organelles.”
(page 17, 671-673)
Comment 47) 679: replace “in a large fraction” with “to a large extent”
Response 47) We replaced “in a large fraction” with “to a large extent”, as suggested.
Comment 48) 725: sentence is incomplete. Did you mean increased glomerular leakage?
Response 48) We now have this mistake and added “leakage”.
Comment 49) 748: mention that the other cell types where APOL1 is expressed are hepatocytes and endothelial cells
Response 49) The review primarily focuses on podocytes. Therefore, unfortunantely the role of other APOL1-expressing cell types could only be addressed to a very limited extent. However, as this point was also raised by Reviewer 2, we have included a sentence with corresponding references in a section on page 18 that now reads:
“Another possible explanation (perhaps in combination with the above-mentioned one) that aligns well with the predominant kidney phenotype, is that podocytes are particularly susceptible to injury mechanisms compared to other cell types suich as hepatocytes and endotheial cells that have been associated with APOL1 RRVs”
(page 18 729-732)

Reviewer 2 Report
Comments and Suggestions for Authors
1. What is the in vivo expression profile of APOL1 splice variants in human kidney, and how does this align with susceptibility to APOL1-mediated cytotoxicity?
2. How do the authors reinterpret the multiple proposed mechanisms of APOL1-mediated toxicity? Can these pathways be sequentially or spatially ordered, or are they context and cell-type dependent?
3. Have any of the key findings (e.g., ER stress, pore formation, lipid interactions) been validated in primary human kidney cells or biopsies from patients with APOL1-associated kidney disease?
4. Can the authors specify what constitutes the “first hit” in podocytes—e.g., interferon-induced signaling, metabolic stress—and how it modulates APOL1 expression, trafficking, or activity?
5. Which of the proposed cytotoxicity mechanisms are most likely to be targeted by emerging therapeutics (e.g., ion channel inhibitors), and how can this inform precision therapy for AMKD?
6. Can the authors provide or cite data comparing the expression thresholds at which wildtype vs. RRV APOL1 becomes cytotoxic, and in which compartments (e.g., ER vs. mitochondria)?
7. How do known genetic modifiers such as APOL3 truncations or UBD expression levels mechanistically interact with the splice variant or membrane topology models presented?
8. Which membrane topology model is best supported by current biochemical or structural data, and what are the implications for APOL1’s intracellular vs. extracellular functions?
Author Response
Point-to-point-response (reviewer 2)
Reviewer 2:
Comment 1) What is the in vivo expression profile of APOL1 splice variants in human kidney, and how does this align with susceptibility to APOL1-mediated cytotoxicity?
Response 1) We agree with the reviewer that, in addition to the presence of RRVs, the expression level and localization of APOL1 - especially its expression profile - are key factors contributing to APOL1-associated diseases. This point was also raised by Reviewer 1 (Reviewer 1; comment 26) in relation to splicing variants. In this context, it is important to note that there is growing evidence suggesting that APOL1-related damage is not limited to renal tissues. For this reason, we have included the review by Pell et al. in our list of references.
The expression profile of APOL1 plays a central role. We have previously shown that APOL1 is expressed at low levels in the brain, and primarily in the liver and kidney (Müller et al., 2021). It is crucial to distinguish not only between organs and tissues (e.g., brain vs. kidney), but also between different cell types (e.g., endothelial cells vs. podocytes), or even between individual cells such as neighboring podocytes, which may show different expression levels, including specific splice variants.
Concerning the question: Most expression studies focus on in vitro or murine systems, the in vivo expression pattern in patients with AMKD is poorly addressed so far, an requires modern techniques such as single-cell transcriptomics (sc or snRNA Seq analyses) to obtain further insights into this matter.
To address the reviewer’s aspect, we included this aspect as a further open question in our manuscript:
“Finally, since most expression studies are based on in vitro or murine models, and the in vivo expression patterns in AMKD patients remain poorly characterized, it is essential to analyze expression profiles at the single-cell level in patient tissues to gain deeper insights into the underlying pathomechanisms.”
(page 7, 257-260)
Comment 2) How do the authors reinterpret the multiple proposed mechanisms of APOL1-mediated toxicity? Can these pathways be sequentially or spatially ordered, or are they context and cell-type dependent?
Response 2) For us, re-interpretation means that future studies must take in vivo expression profiles into account (as noted above, comment 1). However, key cell biological aspects are also crucial, for example, membrane topology, which is a fundamental prerequisite for the various proposed pathomechanisms (see Figs 1-2), as well as the various intracellular APOL1 pools (see Figs 2-5).
There is an increasing and welcomed trend in the APOL1 research field to evaluate the different proposed pathomechanisms not as competing models, but as elements of a spatial-temporal cascade. This integrative approach is currently being explored by several groups (for example, by the groups of Opeyemi Olabisi, Etienne Pays, and many others). It is also possible that these mechanisms may originate from different or independent starting points. An also alternative so for less aspects could be important. For instance: APOL1 may not form a pore or channel itself but might indirectly activate an as-yet unidentified channel, thereby causing ion-imbalances.
In this review, we have aimed to present these aspects in an unbiased and as neutral a manner as possible. The spatial and temporal sequence of pathogenic events is thus a central cell-biological question, but at the same time represents a major experimental challenge. This difficulty arises in part from the strong dependence on the model systems used and the resolution limits of current detection methods.
Comment 3) Have any of the key findings (e.g., ER stress, pore formation, lipid interactions) been validated in primary human kidney cells or biopsies from patients with APOL1-associated kidney disease?
Response 3) This is an excellent point raised by the reviewer, and we fully agree. Although the nephrotoxic effects of APOL1 and its association with kidney diseases have been extensively reported, further studies—particularly in vivo—are needed to elucidate its intracellular and mechanistic functions. Indeed, most existing data are based on in vitro systems or artificial transgenic mouse models (which do not endogenously express APOL1, nor APOL2–4/5).
Current studies using patient material are primarily focused on histological analyses and transcriptomic profiling, which - as mentioned above -(see Comment1) should be expanded to include single-cell approaches. Therefore, key findings require more comprehensive investigation in human specimens derived from AMKD patients. These include APOL1’s subcellular localization and its contribution to various forms of cellular dysfunction. Among these are ER stress, mitochondrial dysfunction, and several forms of programmed cell death such as pyroptosis, apoptosis, and autophagy. In podocytes specifically, APOL1 has also been linked to foot process effacement and detachment from the glomerular basement membrane.
We therefore included the following sentence in the “concluding remarks” paragraph:
Finally, current studies using patient material are primarily focused on histological analyses and transcriptomic profiling. These studies should be expanded to include single cell approaches and high-resolution imaging of human specimens derived from AMKD patients to gain deeper cell biological insights.
(page 19, 782-785)
Comment 4) Can the authors specify what constitutes the “first hit” in podocytes—e.g., interferon-induced signaling, metabolic stress—and how it modulates APOL1 expression, trafficking, or activity?
Response 4) The question of what constitutes the “first hit” in podocytes in AMKD, if APOL1 upregulation on protein level and its downstream effects represents the “second hit,” is both, simple and complex to answer. The simple answer is that any trigger leading to increased APOL1 expression could represent an initial step. In this sense, we agree with the reviewer that interferon-induced signaling cascades (e.g., via the JAK-STAT pathway) or other immunomodulatory stimuli can act as early event and this has been demonstrated in multiple studies. In addition, hypoxia has also been shown to induce high levels of APOL1 expression.
However, the question becomes more difficult when we attempt to understand, at the molecular level, why various diseases - especially those involving podocytopathies -activate signaling pathways that subsequently lead to APOL1 upregulation. In a way, this shifts the question toward identifying the primary molecular and cellular causes of these diseases/podocytopathies. In case of viral infections such as HIV or SARS-CoV-2 (HIVAN and COVAN) the initial trigger is clear. In autoimmune or autoinflammatory diseases, or in the various forms of FSGS, the situation is by far more complex and requires a deeper understanding of which disease-signaling-pathways are broadly associated with AMKD and AMKD-associated podocytopathies. Unraveling these disease-associated signaling pathways is currently the focus of intensive research.
We addressed this important issue by including the following sentences in the manuscript:
„Open questions: While this is an intriguing concept, several open questions remain-including the why various diseases - especially those involving podocytopathies - activate signaling pathways that subsequently lead to APOL1 upregulation. In a way, this shifts the question toward identifying the primary molecular and cellular causes of these podocyte diseases. In case of viral infections, the initial trigger is clear. However, in autoimmune or autoinflammatory diseases, or in the various forms of FSGS, the situation is by far more complex and requires a deeper understanding of which disease-signaling-pathways are broadly associated with AMKD and AMKD-associated podocytopathies.
Furthermore, it is necessary to investigate in more detail how the exact interplay between “pre-damaging” factors and the injury mediated by APOL1 RRVs occurs. The question of why RRVs remain dormant for many years, leading to a relatively late onset of kidney disease, is only partially understood at the cellular level. It is important to identify where the threshold for podocyte injury lies, and which potentially modifiable factors might act as additional elements to delay or prevent the harmful effects of APOL1 RRVs. A precise understanding of these details would also provide a crucial basis for developing therapeutic strategies against AMKDs. “
(page 18, 754-768)
Comment 5) Which of the proposed cytotoxicity mechanisms are most likely to be targeted by emerging therapeutics (e.g., ion channel inhibitors), and how can this inform precision therapy for AMKD?
Response 5) This is another excellent question, which, however, can only be addressed to a limited extent within the scope of this review. There are several strategies for developing druggable targets, many of which are closely linked to the intracellular localization and, consequently, the accessibility of APOL1.
The most straightforward therapeutic approach would be to prevent APOL1 upregulation altogether, for example, by using siRNA-based strategies. Other approaches may aim to directly target specific cytotoxic mechanisms or signaling pathways. However, discussing all these aspects is far beyond the scope of this review.
Increasing evidence suggests that APOL1 may act as a pore or channel at the plasma membrane, or alternatively, may indirectly activate endogenous channels (see Fig. 5). In this context, inhibiting the APOL1 RRV-associated disruption of ion homeostasis represents a particularly direct therapeutic strategy. One example is Inaxaplin (VX-147), which is currently being investigated as a potential ion channel inhibitor targeting APOL1-mediated cytotoxicity.
As mentioned, discussing all these aspects is beyond the scope of this paper, however we addressed the following part to the „open question“ paragraph (see above, comment 4).
Comment 6) Can the authors provide or cite data comparing the expression thresholds at which wildtype vs. RRV APOL1 becomes cytotoxic, and in which compartments (e.g., ER vs. mitochondria)?
Response 6) This concerns a very fundamental question regarding APOL1-mediated toxicity, namely to what extent dose-dependent effects are involved and how APOL1 G0 influences cytotoxicity when co-expressed with the risk variants (RRVs). From clinical data, the issue of a threshold is difficult to investigate, as it is both hard to define and, secondly, patients with AMKD have already exceeded this threshold. There are only a few in vitro studies addressing this specific question; however, a pioneering study by Datta and colleagues (2020) demonstrated a dose-dependent and dominant functionality of the risk variants G1 and G2. The study showed that high levels of APOL1 G0 do not counteract the cytotoxic effects when RRVs are co-expressed, and that the cytotoxicity threshold for RRVs is lower than for APOL1 G0. In our review, we addressed this aspect in section 4.1 “APOL1 cytotoxicity: loss of protection or gain of cytotoxicity?” and cited the corresponding study.
Comment 7) How do known genetic modifiers such as APOL3 truncations or UBD expression levels mechanistically interact with the splice variant or membrane topology models presented?
Response 7) We thank the reviewer for this interesting question. It has been shown that APOL1 and APOL3 physically interact through their C-terminal regions as well as through putative leucine zipper domains in the N-terminal region (Uzureau et al.; Lecordier et al., Skorecki et al.). Recent data further support this interaction by identifying a truncation variant (p.Q58*) that lacks the interaction region and most likely also loses its protective function against APOL1 (Zhang et al.).
Since APOL3 does not contain a SP and is therefore localized in the cytoplasm, the interaction between APOL1 and APOL3 would be restricted to the cytoplasmic APOL1 splice variant vB3 (and possibly vC; see Fig. 2, “like APOL2” topology).
However, the APOL1 splice variant predominantly expressed in the kidney is vA, which follows a different topology and would, once inserted into membranes, no longer be accessible for interaction with cytoplasmic APOL3 (This is as also mentioned at two times in the manuscript text). Nevertheless, it is possible that non-inserted, unprocessed, or misfolded APOL1 vA may still be accessible for interaction. A similar mechanism should be true for UBD (also called FAT10), as only cytoplasmic APOL1 fractions can interact with cytoplasmic UBD and be targeted to the proteasome for inactivation. Again, such an interaction would require the presence of cytoplasmic, and possibly misfolded, APOL1 (or APOL1 vB3 splice variant). The molecular details of these important aspects require further investigations.
Comment 8) Which membrane topology model is best supported by current biochemical or structural data, and what are the implications for APOL1’s intracellular vs. extracellular functions?
Response 8) This is an interesting point: As mentioned above, the aim of this review is to present the various aspects of APOL1 biology as neutrally and unbiased as possible. However, most studies agree that the SP of APOL1 ensures its insertion into the lumen of the ER, from where it is either secreted via the classical secretory pathway or remains as an integral membrane protein in different cellular membranes, mainly at the ER and the plasma membrane.
Several lines of evidence support this, including antibody accessibility studies (Scales et al.), a protein-wide cysteine scanning mutagenesis coupled with a cysteine accessibility assay (Schaub et al.), and the SEAP secretion and N-glycosylation assays from our group (Müller et al.). These findings strongly suggest that APOL1 contains an even number of transmembrane domains, two or four. If APOL1 functions as a pore or channel, it likely consists of four transmembrane helices. The different membrane topologies and their associated pathogenic scenarios are discussed in detail in the review and are summarized in Figures (Figs 1-3).

Round 2
Reviewer 2 Report
Comments and Suggestions for Authors
The paper can be accepted in its present form.